# ShSPI Inhibits Thrombosis Formation and Ischemic Stroke In Vivo

**DOI:** 10.3390/ijms25169003

**Published:** 2024-08-19

**Authors:** Ning Luan, Han Cao, Yunfei Wang, Haihao Zhang, Kangyang Lin, Jingping Hu, Mingqiang Rong, Cunbao Liu

**Affiliations:** 1Institute of Medical Biology, Chinese Academy of Medical Sciences and Peking Union Medical College, Kunming 650118, China; luanning@imbcams.com.cn (N.L.); caohan@imbcams.com.cn (H.C.); wangyf@imbcams.com.cn (Y.W.); zhanghh@imbcams.com.cn (H.Z.); linky6679@163.com (K.L.); hujingping@student.pumc.edu.cn (J.H.); 2National and Local Joint Engineering Laboratory of Animal Peptide Drug Development, College of Life Sciences, Hunan Normal University, Changsha 410081, China

**Keywords:** thrombotic diseases, ShSPI, elastase inhibitor, thrombosis, ischemic stroke

## Abstract

Thrombotic diseases, emerging as a global public health hazard with high mortality and disability rates, pose a significant threat to human health and longevity. Although current antithrombotic therapies are effective in treating these conditions, they often carry a substantial risk of bleeding, highlighting the urgent need for safer therapeutic alternatives. Recent evidence has increasingly pointed to a connection between elastase activity and thrombosis. In the current study, we investigated the antithrombotic effects of ShSPI, an elastase inhibitor peptide derived from the venom of *Scolopendra hainanum*. Results showed that ShSPI significantly attenuated carrageenan-induced thrombosis in vivo. Furthermore, ShSPI effectively inhibited the carrageenan-induced decrease in serum superoxide dismutase (SOD) activity and increase in prothrombin time, fibrinogen level, and endothelial nitric oxide synthase (eNOS) activity. In addition, ShSPI reduced intracerebral thrombosis and improved functional outcomes following ischemic stroke in a transient middle cerebral artery occlusion (tMCAO) mouse model. Collectively, these findings suggest that ShSPI is a promising candidate for the development of novel thrombotic therapies.

## 1. Introduction

The coagulation system, encompassing both the coagulation cascade and platelets, is integral to maintaining hemostasis. However, abnormal activation of this system can lead to thrombosis, a common underlying mechanism involved in thrombotic diseases such as myocardial infarction, ischemic stroke (IS), and venous thromboembolism, collectively representing a leading cause of mortality worldwide [1,2]. Although antiplatelet and anticoagulant therapy, including direct and combined use [3,4], are fundamental in the prevention and treatment of thrombosis, their efficacy is often limited, and the associated risk of bleeding markedly constrains their use [3,5,6]. Thus, there is an urgent need for anticoagulant therapies that do not carry a hemorrhagic risk [7].

In addition to the coagulation cascade and platelets, various molecules contribute to hypercoagulability and thrombus formation [2,8]. Thus, targeting these molecules may inhibit their procoagulant effects without disrupting the overall function of the coagulation system, thereby minimizing the risk of bleeding [9].

Elastases, a broad group of serine proteases that include macrophage elastase, fibroblast elastase, neutrophil elastase, and pancreatic elastase, play a crucial role in the lysis of elastin, the primary component of elastic fibers [10]. The activities of elastases are strictly regulated by several endogenous inhibitors, with an imbalance in this regulation leading to the development of severe diseases [11]. Although elastase is not classified as a coagulation factor, recent studies have implicated elastase in the induction of hypercoagulability and thrombosis. For example, elastase can act on coagulation factors [2] or activate platelets [12], actions that can be inhibited by elastase inhibitors [13,14,15].

In the current study, we investigated the antithrombotic and neuroprotective effects of ShSPI, a specific elastase inhibitor peptide derived from the venom of *Scolopendra hainanum* [16]. Using a carrageenan (CG)-induced mouse tail thrombosis model and transient middle cerebral artery occlusion (tMCAO) mouse model, ShSPI exhibited potent inhibition of thrombosis formation and IS in vivo. These findings suggest that ShSPI holds considerable potential as a candidate for the development of novel therapeutics for thrombotic diseases.

## 2. Results

### 2.1. ShSPI Pretreatment Inhibits CG-Induced Thrombus Formation in an In Vivo Mouse Model

The application of active peptides in vivo is often limited by their potent cytotoxicity and hemolytic activity. To assess the safety and efficacy of ShSPI in vivo, we first evaluated its cytotoxicity against Thp-1 and EA.hy926 cells. As illustrated in Appendix A, ShSPI exhibited no cytotoxicity, even at concentrations as high as 1 mg/mL. We also investigated the hemolytic activity of ShSPI. As shown in Appendix A, ShSPI displayed no significant hemolytic activity at concentrations below 750 μg/mL, and only slight hemolytic activity (6.97%) at a concentration of 1500 μg/mL.

To further explore the antithrombotic potential of ShSPI, we employed a CG-induced mouse tail thrombosis model. As shown in Figure 1, ShSPI treatment significantly inhibited thrombosis formation in a dose-dependent manner. Specifically, following ShSPI administration, thrombosis formation was inhibited by 6.4%, 11.8%, and 32.5% at doses of 0.5, 1, and 2 mg/kg, respectively (Figure 1A). Additionally, thrombus length, expressed as a percentage of mouse tail length, decreased from 85.8% in the CG-induced group to 81.5%, 80.0%, and 55.9% at the respective ShSPI concentrations of 0.5, 1, and 2 mg/kg (Figure 1B). These findings indicate that ShSPI effectively reduces thrombus size and partially inhibits thrombus formation in vivo.

### 2.2. ShSPI Inhibits CG-Induced Coagulation Dysfunction

To assess the impact of ShSPI on coagulation function, we conducted a series of coagulation tests, including prothrombin time (PT), activated partial thromboplastin time (APTT), and fibrinogen levels. As shown in Figure 2A, CG treatment significantly prolonged PT, an effect that was effectively counteracted by ShSPI in a dose-dependent manner. Interestingly, ShSPI treatment did not significantly alter the CG-induced extension of APTT (Figure 2B) but did result in a dose-dependent reduction in CG-induced fibrinogen levels (Figure 2C).

Although elastase is associated with coagulation function [2], it does not serve as a primary coagulation factor within the coagulation cascade. To further elucidate the influence of ShSPI, an elastase inhibitor, on coagulation function, we conducted in vitro experiments. As illustrated in Appendix A, ShSPI had no effect on recalcification time (Appendix A), APTT (Appendix A), or PT (Appendix A). Furthermore, we assessed the impact of ShSPI on the activity of tissue factor (TF), a critical component of the extrinsic coagulation pathway, using a TF activity assay. Consistent with the PT results, ShSPI did not affect TF activity (Appendix A). In summary, the protective effect of ShSPI against CG-induced coagulation dysfunction appears to be independent of direct interaction with the coagulation cascade.

### 2.3. ShSPI Inhibits Platelet Activation

Elastase has been implicated in promoting platelet activation, with platelet hyperactivation identified as a key contributor to thrombosis [17]. To explore the potential inhibitory effects of ShSPI on platelet activation, we administered varying doses of ShSPI (0–4 mg/kg) intravenously (i.v.) via the tail vein. At 2 h post-injection, platelets were isolated and subsequently stimulated with adenosine diphosphate (ADP) to induce aggregation. As shown in Figure 3A,B, ShSPI treatment resulted in a concentration-dependent inhibition of ADP-induced platelet activation. These findings indicate that ShSPI exerts its antithrombotic effects, at least in part, by inhibiting platelet activation, thereby reducing the risk of thrombosis.

### 2.4. ShSPI Modulates Oxidative Stress

Oxidative stress markers, including superoxide dismutase (SOD), glutathione peroxidase (GSH-Px), endothelial nitric oxide synthase (eNOS), and malondialdehyde (MDA), were evaluated in colon tissue from model mice. As depicted in Figure 4, CG treatment led to a significant reduction in SOD (Figure 4A) and GSH-PX (Figure 4B) activities, while eNOS activity was markedly increased (Figure 4D). However, MDA levels remained unchanged following CG induction, with ShSPI treatment also showing no significant influence on MDA levels (Figure 4C). Furthermore, ShSPI administration significantly and dose-dependently attenuated the CG-induced decrease in SOD activity (Figure 4A) and increase in eNOS activity (Figure 4D). Although ShSPI treatment showed a tendency to counteract the CG-induced decrease in GSH-PX activity, the effect was not significant, even at a concentration of 2 mg/kg (Figure 4B). In summary, these findings suggest that ShSPI exerts its antithrombotic effects, at least in part, by modulating oxidative stress pathways.

### 2.5. ShSPI Attenuates IS Severity In Vivo

Elastase inhibitors have been widely recognized for their protective effects against IS [2,10,13]. To evaluate the efficacy of ShSPI in this context, we employed a tMCAO mouse model. As shown in Figure 5A, ShSPI treatment at doses of 0.25 and 1 mg/kg results in a significant, dose-dependent reduction in infarct volumes. Furthermore, this reduction in infarct size correlated with notable improvements in functional outcomes, as evidenced by enhanced performance in the Grip test (Figure 5B), Zea Longa score (Figure 5C), and Bederson score (Figure 5D). These findings collectively indicate that ShSPI offers substantial neuroprotection against IS.

## 3. Discussion

The development of novel antithrombotic drugs with minimal bleeding risks is a critical unmet need. In the present study, we demonstrated that ShSPI, a specific elastase inhibitor, effectively and dose-dependently inhibited thrombus formation in vivo. Moreover, we explored the underlying molecular mechanisms by which ShSPI exerts its antithrombotic effects, highlighting its potential as a therapeutic candidate for treating thrombotic disorders such as IS.

The coagulation system, primarily comprising the coagulation cascade and platelet activity, is central to maintaining hemostasis. In cases of thrombosis driven by the overactivation of this system, current therapeutic strategies typically involve the combined use of coagulation factor inhibitors and platelet inhibitors. Although these therapies are essential for managing thrombotic conditions [18], they are often accompanied by significant bleeding complications, which can lead to disability or even death [19]. Recent research has identified several non-coagulant molecules, including elastase [2], transferrin [9,20], and LL-37 [8,21], as key players in thrombosis via their regulation of the coagulation system. Accordingly, targeted inhibition of these molecules has been shown to reduce hypercoagulability and slow disease progression [9,20,22].

Previous studies have established that excess elastase can promote thrombus formation by activating platelets [23] and enhancing the activity of TF and factor XII (FXII), both of which are integral to the coagulation cascade [2]. Additionally, several elastase inhibitors have demonstrated antithrombotic efficacy [13,24]. We identified ShSPI, a specific elastase inhibitor derived from the venom of *Scolopendra hainanum*, in a previous study [16], which showed potent inhibitory effects on thrombus formation (Figure 1) and IS onset (Figure 5) in our current in vivo mouse model. Furthermore, ShSPI showed no effect on TF activity (Appendix A) but effectively reversed CG-induced prolongation of PT (Figure 2A). These findings suggest that the effects of elastase on TF-dependent coagulation may involve alternative pathways, which warrant further investigation. ShSPI also abolished the CG-induced decrease in SOD activity (Figure 4A) and increase in eNOS activity (Figure 4D) in vivo, although the precise mechanisms by which ShSPI and elastase influence SOD and eNOS require further study.

Resulting from intracerebral vascular occlusion, IS is a leading cause of death and neurological disability worldwide [25]. Currently, the only approved pharmacological treatment for acute IS is intravenous thrombolysis using tissue-type plasminogen activator (tPA), but its effectiveness is limited by the treatment window and hemorrhagic transformation [26]. We found that ShSPI treatment, as a specific inhibitor of elastase, inhibited IS progression, offering a novel treatment approach for managing IS. However, this study primarily focused on the preventive effects of ShSPI against IS, with further research needed to determine the efficacy of elastase inhibitors such as ShSPI when administered at various stages post-IS onset.

In conclusion, our findings indicate that ShSPI exhibits significant inhibitory effects on thrombosis and IS development in vivo, providing a promising treatment candidate for thrombotic diseases, including IS.

## 4. Materials and Methods

### 4.1. Synthesis and Characterization of ShSPI

ShSPI was chemically synthesized by GL Biochem (Shanghai) Ltd. (China) and subsequently refolded using standard protein purification techniques, as described previously [3]. N-terminal sequence analysis confirmed a single protein structure corresponding to CPQVCPAIYQPVFDEFGRMYSNSCEMQRARCLRG, verifying the homogeneity of the isolated inhibitor. The inhibitory activity of ShSPI was validated against human neutrophil elastase (HNE) and porcine pancreatic elastase (PPE) using previously established protocols [16].

### 4.2. Cell Lines and Cytotoxicity Assay

Thp-1 and EA.hy926 cells were cultured in Roswell Park Memorial Institute (RPMI, Thermo Fisher, Grand Island, NY, USA) 1640 medium, supplemented with 10% fetal bovine serum (FBS, Grand Island, NY, USA), 100 U/mL penicillin, and 100 μg/mL streptomycin (Thermo Fisher, Grand Island, NY, USA). All cells were incubated at 37 °C in the presence of 95% air and 5% CO_2_ before use.

For the cytotoxicity assays, Thp-1 cells were seeded in 96-well plates at a concentration of 4 × 10^5^ cells/mL, and treated with 500 nM phorbol 12-myristate 13-acetate (PMA, Beyotime Biotech Inc., Shanghai, China) for 32 h. EA.hy926 cells were seeded at a concentration of 1 × 10^6^ cells/mL. After attachment, cells were washed twice with phosphate-buffered saline (PBS), followed by the addition of RPMI 1640 medium without FBS and overnight incubation. The cell medium was replaced with medium containing double-diluted ShSPI (1000, 500, 250, 125, 62.5, 31.25, 15.63, 7.81, 3.91, and 0 μg/mL). Samples were co-cultured with cells for 18 h, after which the supernatants were discarded, and CCK-8 reagent (MedChemExpress, Monmouth Junction, NJ, USA) was added for an additional 2 h of incubation. Optical density at 450 nm (OD450) was measured to assess cell viability.

### 4.3. Mice

Male ICR mice (4–5 weeks old, 22–25 g) and male C57BL/6 mice (5–6 weeks old, 20–22 g) were obtained from the Animal Center of the Institute of Medical Biology (IMB), Chinese Academy of Medical Sciences (CAMS). All animals were housed in a specific pathogen-free (SPF) environment. All experiments were conducted in accordance with the relevant guidelines established by the Ethics Committee of Animal Care and Welfare of IMB, CAMS, as well as by the Yunnan Provincial Experimental Animal Management Association (permit number: SYXK (dian) K2022-0006) and Animal Ethics Guidelines of the Chinese National Health and Medical Research Council. All efforts were made to minimize animal suffering.

### 4.4. Carrageenan (CG)-Induced Tail Thrombosis Model

To evaluate the antithrombotic properties of ShSPI, a CG-induced tail thrombosis mouse model was utilized. In brief, ShSPI powder was freshly suspended in saline prior to administration. Male ICR mice were randomly divided into six groups: Sham group, mice were administered an intravenous (i.v.) injection of saline (100 μL/mice), followed by an intraperitoneal (i.p.) injection of saline to mimic the modeling process without inducing thrombosis; ShSPI alone group, mice received an i.v. injection of ShSPI (2 mg/kg), followed by an i.p. injection of saline; CG-induced group, mice received an i.v. injection of saline, followed by an i.p. injection of CG (10 g body weight/200 μg CG); CG + ShSPI groups, mice received an i.v. injection of ShSPI at doses of 0.5, 1, or 2 mg/kg body weight, followed by an i.p. injection of CG (10 g body weight/200 μg CG) to induce thrombosis. All mice were maintained in 16 °C during the experimental procedure. The thrombus length and total length of each mouse tail were measured 24 h post-CG injection. After the mice were sacrificed, blood samples were collected into sodium citrate anticoagulant tubes, and colon tissues were harvested for further analysis.

### 4.5. Coagulation Function Analysis

Coagulation function was assessed using whole blood collected from the experimental mice, which was tested using an automatic coagulation analyzer (RAC-1830, Rayto, Shenzhen, China). To determine the effects of ShSPI on coagulation parameters, APTT (R01102, Rayto, Shenzhen, China), PT (R01002, Rayto, Shenzhen, China), FIB concentration (R01302, Rayto, Shenzhen, China), and TF activity (Tissue Factor Activity Assay Kit, ab108906, Abcam, Cambridge, UK) were assayed according to the instructions provided with the respective kits. 

### 4.6. Preparation and Activation of Platelets

C57BL/6 mice were randomly divided into four groups and i.v. injected with varying concentrations of ShSPI (0, 0.25, 1, and 4 mg/kg). At 2 h post-injection, the mice were sacrificed, and whole blood samples were collected into anticoagulation tubes. Platelet-rich plasma (PRP) was isolated from whole blood via centrifugation at 400× *g*/25 °C for 5 min. After washing twice with Tyrode buffer, the inactivated mouse platelets were collected. Platelet aggregation was initiated by the addition of 20 μM ADP, and aggregation curves over 5 min were recorded using an aggregometer (LBY-NJ4, Techlink, Beijing, China).

### 4.7. Transient MCAO (tMCAO)

A tMCAO model was applied to induce focal cerebral ischemia, following previously described methods [20,27]. Male C57BL/6J mice (6 weeks old, 18–20 g) were anesthetized with 3% pentobarbital sodium (80 mg/kg) and fixed in a supine position. An incision was made along the midline of the neck to expose the external and internal carotid arteries. A standardized silicon rubber-coated nylon monofilament (6023910PK10; Doccol, Sharon, MA, USA) was carefully inserted into the left common carotid artery and advanced via the internal carotid artery to occlude the origin of the left middle cerebral artery, thereby inducing ischemia. Reperfusion was initiated by removing the occluding filament 60 min after its insertion. Simultaneously, different concentrations of ShSPI were administered via i.v. injection. After 24 h of modeling, neurological assessments, including Bederson score and Grip test, were performed following previously established protocols [28]. The mice were then sacrificed, and 2 mm thick coronal brain sections were obtained. The sections were stained with 2% 2,3,5-triphenyltetrazolium chloride (TTC, Sigma, Saint Louis, MO, USA) in a dark environment at 37 °C for 10 min, followed by analysis and quantification of infarct volume. To determine the effects of ShSPI on IS, the mice were divided into four groups: Sham Control, Model Control, and two ShSPI groups treated with 0.25 and 1 mg/kg ShSPI, respectively.

### 4.8. Statistical Analyses

All data were analyzed using GraphPad Prism v9.0 (San Diego, CA, USA) and are shown as mean ± standard deviation (SD). For normal continuous variables, one-way analysis of variance (ANOVA) was used. Asterisks represent *p*-value classifications: * *p* < 0.05, ** *p* < 0.01, and *** *p* < 0.001.

## Figures and Tables

**Figure 1 ijms-25-09003-f001:**
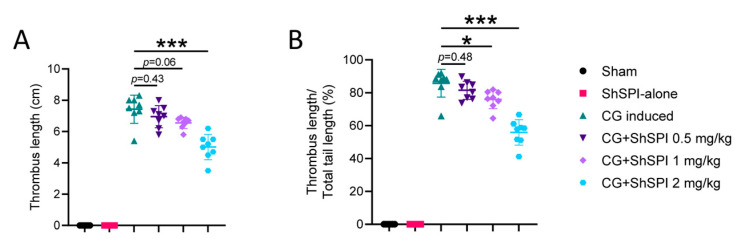
ShSPI inhibited thrombus formation in vivo. (**A**) CG-induced thrombosis in mouse tails was significantly inhibited by ShSPI treatment after 24 h. (**B**) Percentage of thrombus length was also significantly attenuated with ShSPI treatment in a dose-dependent manner. Data are mean ± SD, and compared using one-way analysis of variance followed by Dunnett’s multiple comparisons tests, with group CG induced as control. N = 6 in group Sham and group ShSPI, N = 8 in another four groups, points represent individual mice. CG: carrageenan, * *p* < 0.05, *** *p* < 0.001.

**Figure 2 ijms-25-09003-f002:**
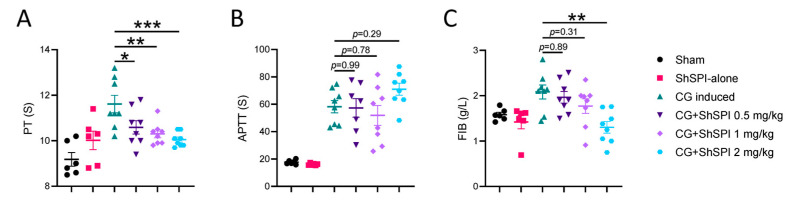
ShSPI inhibited CG-induced coagulation dysfunction. (**A**) CG-induced PT elongation was significantly inhibited by ShSPI treatment at concentrations of 0.5–2 mg/kg. (**B**) ShSPI showed no effects on CG-induced APTT elongation. (**C**) ShSPI significantly inhibited CG-induced FIB elevation in a dose-dependent manner. Data are mean ± SD, and compared using one-way analysis of variance followed by Dunnett’s multiple comparisons tests, with group CG induced as control. N = 6 in group Sham and group ShSPI, N = 8 in another four groups, points represent individual mice. CG: carrageenan, PT: prothrombin time, APTT: activated partial thromboplastin time, FIB: fibrinogen, * *p* < 0.05, ** *p* < 0.01, *** *p* < 0.001.

**Figure 3 ijms-25-09003-f003:**
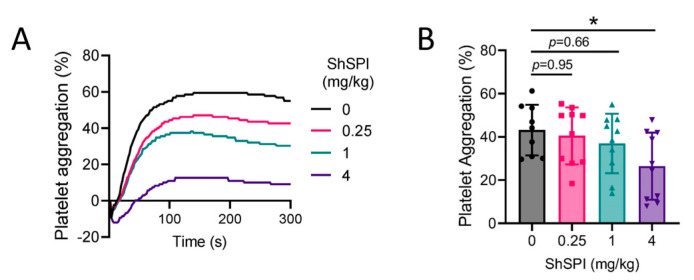
ShSPI inhibits platelet aggregation in a dose-dependent manner. (**A**) Representative images of platelet aggregation of mouse platelet-rich plasma induced by ADP after ShSPI treatment at concentrations of 0-4 mg/kg. (**B**) Statistical analysis of platelet aggregation after ShSPI treatment at concentrations of 0-4 mg/kg. Data are shown as mean ± SD, and compared using one-way analysis of variance followed by Dunnett’s multiple comparisons tests, with ShSPI-0 mg/kg as control, N = 10, and each point represents data from one mouse. * *p* < 0.05.

**Figure 4 ijms-25-09003-f004:**
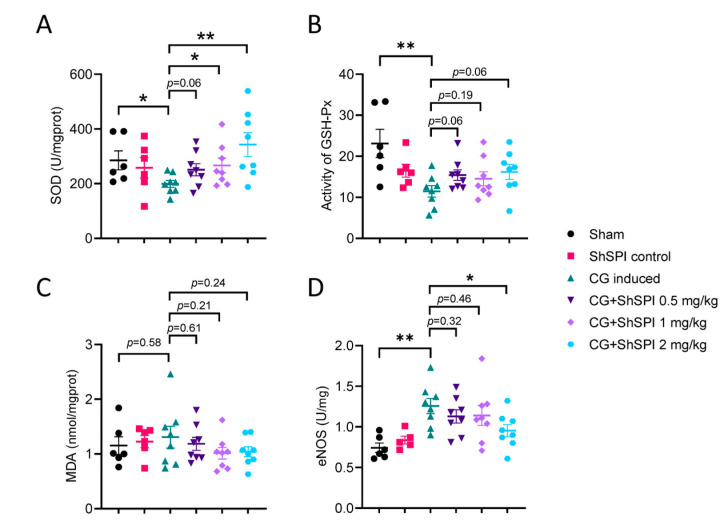
ShSPI modulates oxidative stress. (**A**) ShSPI treatment reversed the CG-induced reduction in SOD activity. (**B**) ShSPI did not significantly inhibit CG-induced decrease in GSH-PX activity. (**C**) CG showed no impact on MDA levels, and ShSPI treatment also showed no effect on MDA. (**D**) ShSPI significantly and dose-dependently abolished the CG-induced increase in eNOS activity. Data are shown as mean ± SD, and compared using one-way analysis of variance followed by Dunnett’s multiple comparisons tests, with group CG induced as control. N = 6 in group Sham and group ShSPI, N = 8 in another four groups, points represent individual mice. CG: carrageenan, SOD: superoxide dismutase, GSH-PX: glutathione peroxidase, MDA: maleicdialdehyde, eNOS: endothelial nitric oxide synthase, * *p* < 0.05, ** *p* < 0.01.

**Figure 5 ijms-25-09003-f005:**
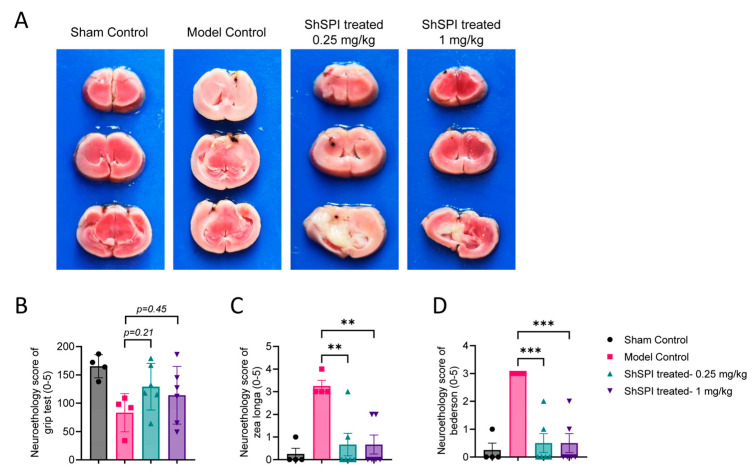
ShSPI protected mice from IS in a tMCAO model. (**A**) Representative brain slices stained with 2% TTC after tMCAO, followed by ShSPI treatment at concentrations of 0.25 and 1 mg/kg. ShSPI treatment significantly improved functional outcomes, as measured by Zea Longa (**C**) and Bederson (**D**) neuroethological scores. While ShSPI treatment tended toward improvement in the Grip test (**B**), the effect was not significant. Data are shown as mean ± SD, and compared using one–way analysis of variance followed by Dunnett’s multiple comparisons tests, with group Model Control as control. N = 4 in group Sham Control and group Model Control, N = 6 in another two ShSPI treated groups, points represent individual mice. ** *p* < 0.01, *** *p* < 0.001.

## Data Availability

All data used during the study are available from the corresponding author upon request.

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
