# Peer review of "ShSPI Inhibits Thrombosis Formation and Ischemic Stroke In Vivo"

_ijms, 2024, doi:10.3390/ijms25169003_

Round 1

Reviewer 1 Report

Comments and Suggestions for Authors

The article presents an intriguing investigation into the antithrombotic effects of ShSPI, an elastase inhibitor peptide derived from the venom of Scolopendra Hainanum. The study explores the potential of ShSPI as a therapeutic agent for thrombotic diseases, addressing a significant public health concern due to the high mortality and disability rates associated with these conditions. While the findings are promising, particularly in the context of novel drug development, there are several aspects of the study that warrant a critical examination, especially regarding the choice of models used.

While carrageenan-induced models are valuable for studying inflammation, their direct relevance to thrombosis, particularly in the context of human thrombotic diseases, is less established. Carrageenan is primarily known for inducing inflammatory responses, which might not accurately mimic the pathophysiological conditions of thrombotic diseases in humans. The use of carrageenan might introduce confounding variables related to its inflammatory effects, which could potentially overshadow the specific thrombotic processes. This may affect the clarity of the results concerning the thrombotic pathways.

Strengths include: The use of ShSPI, derived from centipede venom, introduces a novel approach to antithrombotic therapy. This innovative angle is commendable and opens new avenues for research. The study's dual approach in using both carrageenan-induced thrombosis and transient middle cerebral artery occlusion (tMCAO) models provides a broad perspective on the efficacy of ShSPI in different thrombotic conditions. The investigation into the biochemical effects of ShSPI, such as its impact on serum SOD activity, prothrombin time, fibrinogen level, and eNOS activity, adds depth to the understanding of its mechanism of action.

The background is well-laid, emphasizing the clinical relevance the complex interactions and adverse effects between antiplatelet therapy, anticoagulation therapy, and interventional treatments may also have varying impacts on patient outcomes (see PMID: 36568540 and 32418529). A more detailed examination of the conventional treatments, including their benefits and the associated risks of bleeding, would offer a more complete perspective of the field (for recent reviews, see PMID: 34162232 and PMID: 35976963)

Comments on the Quality of English Language

none

Author Response

We appreciate very much for your professional comments. Minors has been revised as bellows:

  1. Thank you for your comments. While it’s true that the carrageenan model induces thrombosis through the promotion of an inflammatory response, which may not accurately simulate the conditions of thrombotic diseases, it's important to recognize the close relationship between thrombosis and inflammation. Our experimental results demonstrate that ShSPI is effective against carrageenan-induced thrombosis, suggesting that it may have significant therapeutic potential for thrombosis caused by complex factors in the body. Acknowledging the limitations of the carrageenan model, we also evaluated the activity of this peptide in other mouse models.
  2. I’m honored to have your proper advice. I have read references you referred, and made some modification in the revised manuscript (line 34-37).

Reviewer 2 Report

Comments and Suggestions for Authors

In this work the protective effect of a specific elastase inhibitor (ShSPI) against thrombus formation was investigated. The results are interesting and important, as there is a great demand for the development of antithrombotic agents with lower risk of hemorrhagic transformation. However, there are several problems with the presentation of the results:

1.    The number of animals used in the experiments is not indicated anywhere. In the absence of this, statistical power cannot be judged.

2.    For the tMCAO model, when and how were the animals treated with ShSPI?

3.    How was the normality of the data examined during the statistical analysis?

4.    What method was used to determine the fibrinogen concentration?

5.    In the legend of figures 1 and 2, it is stated that "Data are mean +/-SD of at least three independent experiments", while the figure shows individual data points (at least 8 points, very likely the individual results of the treated animals) in each examined group and the horizontal lines presumably indicate the median and interquartile range. The same is true for Figures 3B, 4 and 5B-D. What data are actually shown in the figures?

6.    „2.1. ShSPI prevents carrageenan-induced mouse tail thrombosis in vivo”  and in line 78: „These data suggested that ShSOI can prevent the formation of thrombosis in vivo.”Based on the presented results, pretratment with ShSPI did reduce the size of the thrombus, but it did not completely prevent its formation. 

Minor comments:

7.    Several abbreviations are not explained in the text (line 128: GSH-PX, eNOS; line 131: MDA; line 204: HNE, PPE)

8.    In the legend of Figure 3, plasma-rich platelets is written instead of platelet-rich plasma.

9.    Author Contributions are missing.

Comments on the Quality of English Language

It is recommended to check the quality of the English language.

Author Response

Thank you for your professional comments. Please check our responses to your comments as follows:

  1. Thank you for your prompt, I have added the number of animals in every figure legend which involve with animal experiments.
  2. Animals for the tMCAO model were treated with ShSPI (i.v.) once occluding filament removed. The relevent statement was added in the revised manuscript in line 379-380.
  3. Thanks for your comments, we have not conducted the normal continuous variables analysis, and I have changed the description in the revised manuscript.
  4. APTT, PT, and the concentration of fibrinogen were tested by using automatic coagulation analyzer (RAC-1830, Rayto, China). Method has been listed in the line 353-358 of revised manuscript.
  5. Thank you for your suggestion. All figure Legends have been reformulated.
  6. Thanks for your concern, I have rephrased the statement in the revised manuscript in line 62 (the title of 2.1), and line 77-78 (the conclusion of part 2.1)
  7. Abbreviations like APTT, PT, GSH-PX, eNOS, MDA; HNE, PPE have all been explained in the revised manuscript. Like line 160-162, line 180-182, and line 301-303.
  8. Thank you for your comments. I checked the legend of Figure 3 (line 133 in the revised manuscript) and the whole text, there is no misstatement like “plasma-rich platelets” in this manuscript.
  9. I have attached the “Author Contributions” in the revised manuscript (Line 400-403).